# Association of the Bacteria of the Vermiform Appendix and the Peritoneal Cavity with Complicated Acute Appendicitis in Children

**DOI:** 10.3390/diagnostics13111839

**Published:** 2023-05-24

**Authors:** Konstantinos Zachos, Fevronia Kolonitsiou, Antonios Panagidis, Despoina Gkentzi, Sotirios Fouzas, Vasileios Alexopoulos, Eirini Kostopoulou, Stylianos Roupakias, Aggeliki Vervenioti, Theodore Dassios, George Georgiou, Xenophon Sinopidis

**Affiliations:** 1Department of Pediatric Surgery, Children’s Hospital, 26331 Patras, Greece; zachos.i.k@gmail.com (K.Z.);; 2Department of Microbiology, University of Patras School of Medicine, 26504 Patras, Greece; 3Department of Pediatrics, University of Patras School of Medicine, 26504 Patras, Greece; 4Department of Pediatric Surgery, University of Patras School of Medicine, 26504 Patras, Greece

**Keywords:** acute appendicitis, bacterial culture, children, appendiceal lumen, peritoneal cavity, complicated appendicitis

## Abstract

Background: Primary infection has been questioned as the pathogenetic cause of acute appendicitis. We attempted to identify the bacteria involved and to investigate if their species, types, or combinations affected the severity of acute appendicitis in children. Methods: Samples from both the appendiceal lumen and the peritoneal cavity of 72 children who underwent appendectomy were collected to perform bacterial culture analysis. The outcomes were studied to identify if and how they were associated with the severity of the disease. Regression analysis was performed to identify any risk factors associated with complicated appendicitis. Results: *Escherichia coli*, *Pseudomonas aeruginosa*, and *Streptococcus species* were the most common pathogens found in the study population. The same microorganisms, either combined or separate, were the most common in the appendiceal lumen and the peritoneal cavity of patients with complicated appendicitis. Gram-negative bacteria and polymicrobial cultures in the peritoneal fluid and in the appendiceal lumen were associated with complicated appendicitis. Polymicrobial cultures in the peritoneal cavity presented a four times higher risk of complicated appendicitis. Conclusions: Polymicrobial presentation and Gram-negative bacteria are associated with complicated appendicitis. Antibiotic regimens should target the combinations of the most frequently identified pathogens, speculating the value of early antipseudomonal intervention.

## 1. Introduction

The most common cause of acute abdominal pain requiring surgery is acute appendicitis. In the developed world, a total of 5.7–50 per 100,000 inhabitants is affected by acute appendicitis every year [1]. More than 30% of the patients are children with a peak incidence at 11–12 years of age [2]. Children younger than five years often present atypical symptoms with a complication rate of up to 50% [3,4,5,6]. The fact that there are still children with peritonitis in the era of highly accurate diagnostic modalities constitutes a challenge for the present generation of clinicians and researchers [7]. The incidence of complicated appendicitis may reach 100% in children under the age of three years [8,9,10]. Early detection and administration of antibiotics are essential, as timely treatment affects prognosis [11].

Though acute appendicitis has been studied extensively, its pathogenesis is still a matter of debate. Obstruction of the appendiceal lumen has been traditionally considered the trigger of inflammation [12,13]. However, the theory that acute appendicitis is caused by primary bacterial infection with secondary luminal obstruction has been gaining ground. This is supported by evidence that acute appendicitis occurs in clusters, presents seasonal variation, and is less common in rural areas of the developing world [14,15].

During the last two decades, there is a trend to consider acute appendicitis as a double entity: a disease characterized as uncomplicated, which may regress under certain circumstances with or without the help of antibiotics, and a disease characterized as complicated, which should be treated timely with open or laparoscopic appendectomy [16,17]. It is known that not only uncomplicated but occasionally complicated appendicitis, such as abscess or phlegmon, can be initially treated safely and effectively with antibiotics [16,17]. In both types, the intervention of any antibiotic involved should be focused on the appropriate bacteria. Therefore, it is considered essential to study the microbiome of the appendix and the peritoneal cavity of patients with acute appendicitis, especially children [18].

In the present study, we aimed to describe the bacterial profile of the appendiceal lumen and the peritoneal cavity in a cohort of children with acute appendicitis using a culture-based approach. Furthermore, we aimed to investigate any possible association between the culture outcomes and the severity of appendicitis by dividing the cohort into patients with uncomplicated and complicated appendicitis, targeting the accomplishment of a most effective anti-microbial regimen.

## 2. Materials and Methods

### 2.1. Study Population

This is a prospective study on a cohort of children with acute appendicitis who underwent appendectomy with laparoscopy or open surgery during a period of one year. Medical history, clinical examination, and routine laboratory tests were recorded on admission. Patients with chronic disorders (respiratory, cardiovascular, renal, metabolic, etc.), hematological diseases or malignancies, and those who had received antibiotics prior to admission were not included in the study. Children younger than four years and those without postoperative histopathological evidence of acute appendicitis were excluded as well. Cefuroxime and metronidazole were administrated according to the department’s protocol for a maximum period of 18 h prior to surgery. Amikacin was added if the anticipated diagnosis was complicated appendicitis according to the clinical presentation and the preoperative ultrasound findings. The antibiotic regimen was adjusted postoperatively according to the intraoperative findings. The initial macroscopic assessment of the stage of appendicitis during surgery was confirmed by histopathology.

### 2.2. Sample Collection

After inspecting the abdominal cavity and prior to any manipulation, a sample of peritoneal fluid was obtained with aspiration through a sterilized feeding tube. The excised appendix was washed under sterile conditions with normal saline 0.9%. The material from the appendiceal lumen was extracted with light milking into a sterilized plastic sample cup. Finally, the appendix was placed into formalin for histopathological analysis.

### 2.3. Histopathology

The laparoscopic grading system was adopted to define the appendicitis stage [16]. The score includes normal appendix (grade 0), hyperemia and oedema (grade 1), fibrinous exudate (grade 2), segmental necrosis (grade 3a), necrosis at the base of the appendix (grade 3b), abscess (grade 4a), regional peritonitis (grade 4b), and diffuse peritonitis (grade 5) [19,20]. Grades 1–2 are characterized as non-complicated appendicitis, while grades 3–5 are characterized as complicated [19,21]. Patients with grade 0 were not included in the study.

### 2.4. Microbiological Assessment

The fluid from the peritoneal cavity was injected into blood culture bottles (BACT/ALERT FA Plus, BACT/ALERT FN Plus-bioMérieux, Craponne, France) immediately after sampling and, together with the material from the lumen, were transferred to the microbiology department. Once arrived, the sample from the sample cup was diluted with normal saline 0.9% and inoculated in MacConkey agar plates and Schaedler agar plates with 5% horse blood (Oxoid Ltd., Dublin, Ireland). The plates were incubated in aerobic and anaerobic atmospheres at 37 °C, and bacterial growth was examined after 24 and 48 h of incubation. All bacteria isolated in aerobic and anaerobic conditions were identified with the VITEK 2 system (bioMérieux, France) and the API 20A system (bioMérieux, France). Antimicrobial susceptibility testing was performed with the automated VITEK-2 system (bioMérieux, France) and the ETEST method (bioMérieux, France) according to the principles of the European committee on antimicrobial susceptibility testing (EUCAST).

Particular attention was given to the direct inoculation of the clinical samples for the survival of oxygen-sensitive microorganisms such as anaerobes. The culture media facilitated the growth of fastidious as well as anaerobic microorganisms. Based on the current situation, the molecular-based mass spectrometric identification method enabled a faster and more efficient identification of microorganisms to the species level and, in many cases, detected some enzymes, such as β-lactamase, offering an alternative approach to a phenotypic resistance testing.

According to the culture outcomes, the bacteria were described as single units or as groups and were clustered as aerobic, facultative, and obligatory anaerobic; Gram-positive; and Gram-negative. The cultures were categorized as monomicrobial and polymicrobial according to the growth of one and at least two bacteria, respectively. Patients with negative cultures of the lumen of the appendix were excluded from the study.

### 2.5. Statistical Analysis

For the data analysis, the potential effect of the independent variables (outcomes from the cultures of the lumen of the appendix and the peritoneal cavity) on the dependent variables (uncomplicated and complicated acute appendicitis) was examined using descriptive statistics of the correlation coefficient. In some parts of the statistical analysis, the correlation between a qualitative and a quantitative variable was tested. For this reason, the t-Kendall was used as the optimal correlation coefficient, which converges faster to the normal distribution than Spearman’s correlation coefficient because of the rank orders. For the rest of the analysis, the chi-square test was employed with the respective measures of correlation (phi coefficient, Cramer’s V, contingency coefficient) to determine the correlation between qualitative independent and dependent variables. For the analysis of the possible correlation between the culture outcomes and the severity of appendicitis, a logistic regression model was used (Logit regression). All statistical analyses were performed using the SPSS Statistical Software Package version 25 (IBM Corp., Armonk, NY, USA). The threshold for statistical significance was defined as *p* < 0.05.

### 2.6. Ethical Considerations

The study was conducted in accordance with the Declaration of Helsinki and approved by the Ethics Committee of the Patras Children’s Hospital (2619/04 March 2019). Informed consent was obtained from the parents of the patients prior to their enrolment in the study.

## 3. Results

A cohort of 72 children (43 boys and 29 girls) who were submitted to appendectomy from 1st January to 31st December, 2020 were enlisted in the study. The mean age of the study population was 10.6 years (range: 6–16 years). The demographic characteristics, clinical presentation, and laboratory findings are presented in Table 1. There were 42 children (58.3%) with uncomplicated appendicitis and 30 children (41.7%) with complicated appendicitis (Table 1).

A total of 8.30% of the specimens presented histopathologic findings of hyperemia and oedema (grade I), and 50% of the specimens presented histopathologic findings of fibrinous exudate (grade II), both considered uncomplicated appendicitis. Complicated appendicitis was represented by 9.8% of the specimens with base necrosis (grade 3a), 16.6% with abscess (grade 4a), 12.5% with local peritonitis (grade 4b), and 2.8% with general peritonitis (grade 5) (Figure 1).

### 3.1. Appendiceal Lumen Outcomes

The cultures of the lumen of the appendix showed that 73.6% of the isolated bacteria of the study population were Gram-negative, and 23.4% were a mixed population of Gram-positive and Gram-negative microorganisms. No culture included Gram-positive bacteria exclusively.

Aerobic and facultative anaerobic bacteria were developed in the lumen in 82% of cultures, obligatory aerobic in 2.7%, and combined in 15.3%. Polymicrobial cultures were 55.5%, and monomicrobial cultures were 45.5%.

*Escherichia coli* was the most commonly identified microorganism in the appendiceal lumen (83.4%; 34.7% as a single entity) followed by *Pseudomonas aeruginosa* (19.5%), *Streptococcus species* (spp.) (12.5%), *Enterococcus faecalis* (11.1%) that was always found in association with other bacteria, and *Klebsiella pneumoniae* (8.3%) (Table 2).

*E. coli* was found in 76.2% of the cultures of the appendiceal lumen in patients with uncomplicated appendicitis (35.7% as the only pathogen) followed by *P. aeruginosa* (21.5%) and *K. pneumoniae* (14.3%) (Table 3).

*E. coli* was also the predominant microorganism in complicated appendicitis (96.4%; 63.1% of them in combination with other bacteria). The next most common pathogens were *Streptococcus* spp. (20%) and *P. aeruginosa* (16.6%) (Table 3).

### 3.2. Peritoneal Cavity Outcomes

A total of 45% of the cultures from the peritoneal cavity were positive for bacteria. The percentages that follow refer exclusively to the positive culture outcomes.

Gram-negative bacteria were developed in 62.5% of the cultures, Gram-positive bacteria were developed in 31.2% of the cultures, and a combination of Gram-positive and Gram-negative bacteria were developed in 6.3% of the cultures.

In all of these cultures except for one, all microorganisms were aerobic and facultative anaerobic. A total of 59.4% of the cultures of the peritoneal fluid were monomicrobial, and 41.5% were polymicrobial.

*E. coli* was the most common pathogen (77.4%; 45.2% as monomicrobial culture) followed by *Streptococcus* spp. (19.3%) and *P. aeruginosa* (9.2%; always combined with other bacteria) (Table 4).

In the uncomplicated cases, *E. coli* was the most commonly identified bacterium (54.6%; 45.5% as the only pathogen of the culture), while *K. pneumoniae* and *E. faecalis* shared an equal percentage of 18.2% (Table 5).

In patients with complicated appendicitis, *E. coli* (90%; 45% as a monomicrobial culture) and *Streptococcus* spp. (30%; 5% as monomicrobial) were the most frequent bacteria encountered (Table 5). *P. aeruginosa* was found in 10% combined with *E. coli*, while *Pseudomonas vesicularis* was found in 5% combined with *Streptococcus* spp. (Table 5).

### 3.3. Interpretation of the Culture Outcomes

When a possible association between the culture outcomes and the severity of appendicitis was questioned in regard to the bacterial species, none was found statistically significant, both in the appendiceal lumen and the peritoneal cavity.

Nevertheless, when the bacteria were categorized into groups, Gram-negative bacteria (*p* = 0.003), polymicrobial cultures (*p* < 0.01) in the peritoneal fluid, and polymicrobial cultures in the lumen of the appendix (*p* = 0.04) were found to be associated with complicated appendicitis.

With the logistic regression analysis, we tried to determine if the correlations that were displayed herein with conventional statistical methods were valid towards a certain direction, i.e., if the different groupings of the culture outcomes affected the presence of uncomplicated or complicated appendicitis. With this method, we created multiple models from the combinations of the formed groups, and we evaluated their possible association with the dependent variables through Wald testing. The coefficient of each model corresponded to a particular *p*-value, resulting in the significance or not of the way an independent variable affected the dependent one.

Among all combinations, the model with the value (2) according to the Wald test was the only one that presented statistical significance with a *p*-value lower than 0.05 (Table 6). For the needs of analysis, the value (1) was given to the model of the non-complicated cases of acute appendicitis, and the value (2) was given to the model of the complicated ones (as the dependent variable). Accordingly, the value (1) was given to the cultures with a single microorganism, and to those with two or more microorganisms, the value (2) (independent variables) was given. The outcome that resulted was that, when there was more than one species of bacteria in the culture of the peritoneal fluid (polymicrobial culture), there was significant risk of presentation of complicated appendicitis (Table 6).

Thus, it was shown that polymicrobial cultures in the peritoneal fluid were correlated with a four times higher risk of complicated appendicitis (Odds ratio: 3.88, 95% confidence interval: 1.82–8.25) (Table 6).

## 4. Discussion

In this study, we attempted to identify the bacteria of the appendiceal lumen and the peritoneal cavity in a cohort of children with acute appendicitis and to find any association with the severity of inflammation and the risk of complications. Similar studies have been performed that intended to clarify the pathophysiology of the disease and to create protocols of therapy [10,12,13].

We live in the era of a shifting trend from immediate appendectomy in all cases with acute appendicitis to watchful observation regarding non-complicated appendicitis and delayed treatment either with surgery if the clinical presentation worsens or with conservative management with antibiotics [22,23,24]. The vermiform appendix is no longer considered a redundant, useless apparatus of the gastrointestinal tube [25,26]. In opposite, it is considered a reservoir of a biofilm of bacteria, which colonizes the caecum and the large intestine and provides protection against pathogenic microorganisms [27,28]. A gene sequencing study proved the development of fungi in a healthy population with appendectomy in contrast to controls who kept their appendix [29].

According to international guidelines, delayed appendectomy until the first 24 h after the presentation of symptoms is safe in most cases and does not increase the risk of complications [26,30]. According to the recent international guidelines on appendicitis, complicated cases of diffuse peritonitis should be treated with surgery no more than 8 h after diagnosis, while the surgical management of special forms such as phlegmons may be anticipated [17,31].

The recent restriction measures during the COVID-19 pandemic and their impact on the incidence and the treatment strategies of acute appendicitis, especially on the use of antibiotics during this period, played a critical role in the conservative management. This resulted in a stricter selection for operative treatment and a longer waiting time between the onset of symptoms and intervention. This policy continued as a clinical practice after the end of the pandemic [32,33,34].

The current management of acute appendicitis renders the selection of the antibiotic regimen extremely important for an optimal effect on the pathogenic factors and achievement of a better possible outcome [35]. Therefore, knowledge of the most common bacteria that are implicated in this process is of great value. The bacterial flora of the appendix and/or the peritoneal cavity in children with acute appendicitis has been described in numerous studies [20,36,37,38,39,40,41,42,43,44,45,46,47,48]. The association between certain bacteria and the stage of appendicitis has been investigated as well [20,36,37,41,44,45,47,48].

Traditional culture techniques [20,36,37,38,39,40,41,42], culture-independent methods such as rRNA or rDNA gene sequencing [43,44,45,46,47], or both [48] have been used to investigate the microbial content and its role in the different grades of appendicitis. Supporters of the gene sequencing studies reported that traditional cultures failed to identify the majority of the microorganisms [41], as children with acute appendicitis presented an abundance of the genera *Fusobacteria* and a decrease in the genera *Bacteroides* compared to controls [44,46]. These studies reported an association with the severity of inflammation, but they did not reach statistical significance and did not assign causality [44,46].

Based on the theory that gene sequencing techniques cannot distinguish between viable and non-viable bacteria, other studies preferred traditional cultures after collecting fluid from the peritoneal cavity [20,36,37,38,39,40,41,42]. Schülin et al. collected samples from the lumen of the appendix [48]. This study was the only one to our knowledge in which both traditional cultures and genetic material sequencing were simultaneously used [48].

*E. coli* is reported as the most common microorganism followed by *P. aeruginosa*, *Bacteroides* spp., and *Streptococcus* spp. [30,36,37,38,39,40,41,42,48]. No statistically significant association has ever been reported between specific bacterial strains and the different stages of appendicitis, although in some studies, *P. aeruginosa* was often identified in gangrenous and the most severe forms of appendicitis [20,46]. An increase in the *Streptococcus milleri* group has been reported in complicated cases as well [41,49].

Our results agreed with those of most of the studies with *E. Coli*, *P. aeruginosa*, and *Streptococcus* spp. being the most frequent pathogens [20,36,37,38,39,40,41,42,50]. In perforated appendicitis, *Streptococcus* spp. and *P. aeruginosa* were also common in the lumen, while *Streptococcus* spp. and *P. aeruginosa* were common in the peritoneal cavity. However, in our study population, it was not a certain species but the categories and combinations of bacteria that presented a statistically significant association with severe inflammation.

Given the high prevalence of *P. aeruginosa*, isolated in the cases of complicated appendicitis, antibiotic regimens with antipseudomonal activity in early clinical stages could potentially be considered more intensively in the future [51,52]. Furthermore, as the conservative approach is currently under discussion for uncomplicated appendicitis, well designed studies may perhaps address this issue and clarify any possible effect of regimens, including anti-pseudomonas coverage, on the outcome of such cases versus standard antimicrobial care.

More recent studies comparing bacterial cultures between patients with perforated and non-perforated appendicitis reported analogous outcomes. *E. coli* was the most common microorganism both in acute and perforated appendicitis irrespective of the clinical presentation, a finding encountered in our study population as well [53,54]. Though non-correlated directly with complications, it presented extended spectrum β-lactamase production [53]. Tamura et al. showed domination of *E. coli* and *Bacteroides* species. *P. aeruginosa*, *Streptococcus anginosus*, and *Enterococcus* groups were significantly higher in complicated appendicitis [55]. *P. aeruginosa* was also encountered in other studies in complicated forms of appendicitis as in our study with more strains presenting resistance to a number of antibiotics [53]. It is of interest that, in a very recent study, *Fusobacterium* was found in high concentrations in the appendiceal lumen but also in the saliva and feces of children with acute appendicitis compared to the normal population, implying an important role of ectopic colonization of *Fusobacterium* in the process of appendicitis [56].

Equilibrium between antibiotic effectiveness and protection from bacterial resistance to antibiotics should infiltrate the philosophy of every clinical decision on the use of these drugs against microorganisms implicated in appendicitis. Taking into consideration this perspective, as more than one bacterium were present in serious forms of appendicitis in our study, a targeted combination of antibiotics should be used prior to surgery in patients with a clinical presentation implying complicated appendicitis. Although it is clear that appendectomy should be performed in diffused peritonitis, there is debate on primary appendectomy in patients with abscess or plastron [18]. In these patients, the administration of combined antibiotics is crucial. To avoid overtreatment and resistance, culture-based antibiotic therapy modifications are required postoperatively, especially shifting from triple to double regimens when mandatory.

This study has certain limitations. We must also take into consideration that the age selection criteria resulted in outcomes representative of the study age group, as younger children who represent an age group with higher prevalence of complicated appendicitis were excluded. The study population is limited, and the nature of the study is a prospective cohort instead of a randomized controlled one. Thus, further research is required to confirm the sample outcomes in a larger population, ideally through a multicenter approach.

## 5. Conclusions

*E. coli*, *P. aeruginosa*, and *Streptococcus* spp. are the most common microorganisms in children with acute appendicitis. However, the presence of more than one bacterium, both in the lumen of the appendix and in the peritoneal cavity, seems to be the most important factor associated with the severity of inflammation and the presence of complications. Maybe a possible synergic collaboration of microorganisms affects the expression of appendicitis gravity. As non-invasive treatment of acute appendicitis gains more ground lately, antibiotic treatment should be targeted to combinations of the most often identified pathogens. In particular, the application of regimens including antipseudomonal agents in a more routine way may become a game changer and might be considered for the early phases of the management.

Regarding the younger physicians who enter the field and the struggle against appendicitis in children, they should bear in mind certain issues: a. The most commonly involved bacteria in acute complicated appendicitis include *E. coli*, *P. aeruginosa*, and *Enterococcus* species. Therefore, antibiotic regimens should cover these types, especially in more serious clinical forms, as a principle. b. We live in the era of conservative treatment of acute simple appendicitis as an option with antibiotic therapy being our main tool. However, as the bacteria develop resistance to antibiotics, their use should be targeted and judicious. c. Interaction between specialties (pediatric surgeons, pediatricians, radiologists, microbiologists, infectious disease specialists, molecular biologists) should be a prerequisite if we want to improve our intervention against bacteria. The era of the single deciding physician who is considered as the authority should become a thing of the past. d. Last but not least, medical perceptions are modified daily. We should anticipate and confront any changes ahead of them instead of following their evolution.

## Figures and Tables

**Figure 1 diagnostics-13-01839-f001:**
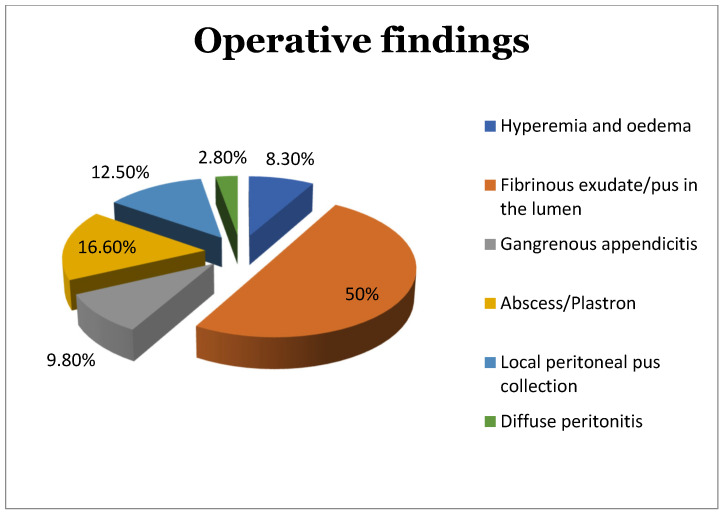
Operative histopathologic findings of the study population.

**Table 1 diagnostics-13-01839-t001:** Demographic, clinical, and laboratory outcomes of the study population.

Variables	Uncomplicated Appendicitis	Complicated Appendicitis	*p*-Value
*n*	42	30	
Female	16	13	0.444
Male	26	17	0.069
Age (years)	10.8 ± 2.4	10.4 ± 3.0	0.449
Weight (kg)	41.5 ± 13.4	41.2 ± 15.7	0.932
Height (cm)	151.5 ± 13.7	148.7 ± 19	0.467
BMI (kg/m^2^)	17.6 ± 3.6	17.9 ± 3.2	0.515
RLQ tenderness	41 (97.6%)	30 (100%)	0.583
Excessive RLQ tenderness	21 (50%)	19 (63.3%)	0.262
Pain migration	18 (42.9%)	14 (46.7%)	0.748
Anorexia	31 (73.8%)	25 (83.3%)	0.338
Nausea/emesis	28 (66.7%)	27 (90%)	0.022
Temperature max (°C)	37.3 ± 0.8	38.0 ± 0.7	<0.001
WBC (10^3^/μL) *	14.0 ± 3.5	16.0 ± 4.2	0.017
Neutrophil count (10^3^/μL) *	11 ± 3.6	13.5 ± 3.9	0.007
Neutrophils (%) *	77.6 ± 8.8	83.6 ± 4.6	<0.001
Hb (g/dL) *	12.96 ± 1.02	12.93 ± 0.18	0.685
CRP (mg/dL) *	3.4 ± 4.6	6.9 ± 5.9	0.015

Abbreviations: BMI: Body mass index, RLQ: Right lower quadrant, WBC: White blood cells, CRP: C-reactive protein. * Laboratory normal values: WBC (10^3^/mL): 3.6–10.0, neutrophil count (10^3^/mL): 1.5–7.5, neutrophils (%): 45.0–75.0, Hb (g/dL): 12–18, CRP (mg/dL): ≤ 0.6.

**Table 2 diagnostics-13-01839-t002:** Bacteria and their combinations as they were identified in the lumen of the appendix of the study population.

Bacteria	%
*Escherichia coli*	34.7
*Escherichia coli + Pseudomonas aeruginosa*	12.5
*Escherichia coli + Streptococcus* spp.	9.7
*Escherichia coli + Enterococcus faecalis*	8.3
*Klebsiella pneumoniae*	6.9
*Escherichia coli + Bacteroides* spp. *(non-fragilis)*	5.6
*Pseudomonas aeruginosa*	2.8
*Escherichia coli + Bacteroides fragilis*	2.8
*Bacteroides fragilis*	1.4
*Escherichia coli + Klebsiella pneumonia*	1.4
*Escherichia coli + Enterococcus avium*	1.4
*Escherichia coli + Enterococcus gallinarum*	1.4
*Escherichia coli + Proteus mirabilis*	1.4
*Streptococcus* spp. *+ Bacteroides* spp.	1.4
*Escherichia coli + Propionebacterium* spp.	1.4
*Escherichia coli + Pseudomonas aeruginosa + Enterococcus faecalis*	1.4
*Escherichia coli + Providencia rettgeri + Clostridium* spp.	1.4
*Pseudomonas aeruginosa + Providencia rettgeri + Bacteroides ovatus*	1.4
*Pseudomonas aeruginosa + Klebsiella pneumoniae + Enterococcus faecalis*	1.4
*Escherichia coli + Streptococcus* spp. *+ Βacteroides* spp. *+ Enterobacter aerogenes*	1.4

**Table 3 diagnostics-13-01839-t003:** Bacteria identified in the lumen of the appendix in uncomplicated and complicated appendicitis.

Bacteria	%
Uncomplicated appendicitis	
*Escherichia coli*	35.7
*Klebsiella pneumoniae*	11.9
*Escherichia coli + Pseudomonas aeruginosa*	11.9
*Escherichia coli + Enterococcus faecalis*	9.5
*Escherichia coli + Bacteroides* spp. *(non-fragilis)*	7.1
*Pseudomonas aeruginosa*	4.8
*Escherichia coli + Streptococcus* spp.	4.8
*Bacteroides fragilis*	2.4
*Escherichia coli + Enterococcus gallinarum*	2.4
*Escherichia coli + Bacteroides fragilis*	2.4
*Pseudomonas aeruginosa + Pseudomonas rettgeri + Bacteroides ovatus*	2.4
*Pseudomonas aeruginosa + Klebsiella pneumoniae + Enterococcus faecalis*	2.4
*Escherichia coli + Streptococcus* spp. *+ Bacteroides* spp. *(non-fragilis) + Enterobacter aerogenes*	2.4
Complicated appendicitis	
*Escherichia coli*	33.3
*Escherichia coli + Streptococcus* spp.	16.7
*Escherichia coli + Pseudomonas aeruginosa*	13.3
*Escherichia coli + Enterococcus faecalis*	6.7
*Escherichia coli + Klebsiella pneumoniae*	3.3
*Escherichia coli + Bacteroides* spp. *(non-fragilis)*	3.3
*Escherichia coli + Enterococcus avium*	3.3
*Escherichia coli + Enterococcus gallinarum*	3.3
*Escherichia coli + Bacteroides fragilis*	3.3
*Streptococcus* spp. *+ Bacteroides* spp.	3.3
*Escherichia coli + Propionebacterium* spp.	3.3
*Escherichia coli + Pseudomonas aeruginosa + Enterococcus faecalis*	3.3
*Escherichia coli + Providencia rettgeri + Clostridium* spp.	3.3

**Table 4 diagnostics-13-01839-t004:** Bacteria and their combinations as they were identified in the peritoneal cavity of the study population.

Bacteria	%
*Escherichia coli*	45.2
*Escherichia coli + Streptococcus* spp.	12.9
*Escherichia coli + Pseudomonas aeruginosa*	9.7
*Enterococcus faecalis*	3.2
*Bacteroides fragilis*	3.2
*Klebsiella pneumoniae*	3.2
*Providencia rettgeri*	3.2
*Streptococcus* spp.	3.2
*Escherichia coli + Enterococcus faecalis*	3.2
*Escherichia coli + Enterococcus avium*	3.2
*Escherichia coli + Enterococcus gallinarum*	3.2
*Streptococcus* spp. *+ Pseudomonas vesicularis*	3.2
*Klebsiella pneumoniae + Enterococcus faecalis*	3.2

**Table 5 diagnostics-13-01839-t005:** Bacteria identified in the peritoneal cavity in uncomplicated and complicated appendicitis.

Bacteria	%
Uncomplicated appendicitis	
*Escherichia coli*	45.5
*Enterococcus faecalis*	9.1
*Bacteroides fragilis*	9.1
*Klebsiella pneumoniae*	9.1
*Providencia rettgeri*	9.1
*Escherichia coli + Pseudomonas aeruginosa*	9.1
*Enterococcus faecalis + Klebsiella pneumoniae*	9.1
Complicated appendicitis	
*Escherichia coli*	45
*Escherichia coli + Streptococcus* spp.	20
*Escherichia coli + Pseudomonas aeruginosa*	10
*Streptococcus* spp.	5
*Escherichia coli + Enterococcus faecalis*	5
*Escherichia coli + Enterococcus avium*	5
*Escherichia coli + Enterococcus gallinarum*	5
*Streptococcus* spp. *+ Pseudomonas vesicularis*	5

**Table 6 diagnostics-13-01839-t006:** Logistic regression analysis tested the correlation of peritoneal fluid and appendiceal lumen cultures of patients with complicated appendicitis with monomicrobial and polymicrobial cultures. Statistically significant outcomes are shown in bold characters.

Variable	Model Value	Number of Patients			
Operative findings	Complicated appendicitis (2)	30			
	Uncomplicated appendicitis (1)	42			
	Total	72			
Prediction coefficient	Coef.	SE Coef.	Z	*p*-value	Odds Ratio
Variable	−1.39208	1.57541	−0.88	0.377	
**Peritoneal fluid culture (2)**	**1.35473**	**0.385695**	**3.51**	**0.001**	**3.88**
Appendiceal lumen culture (2)	0.658258	0.544083	1.21	0.226	1.93
95% CI					
Prediction coefficient	Lower	Upper			
Variable					
**Peritoneal fluid culture (2)**	**1.82**	**8.25**			
Appendiceal lumen culture (2)	0.66	5.61			
Log-Likelihood = −41.802					
Test that all slopes are zero: G = 22.504, DF = 4, *p*-value = 0.000					
Goodness-of-fit tests					
Method	Chi square	DF	*p*-value		
Pearson	64.9860	56	0.192		
Deviance	71.4668	56	0.080		
Hosmer–Lemeshow	6.9920	8	0.537		

## Data Availability

Data is unavailable due to privacy.

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
