# Peer review of "Association of the Bacteria of the Vermiform Appendix and the Peritoneal Cavity with Complicated Acute Appendicitis in Children"

_diagnostics, 2023, doi:10.3390/diagnostics13111839_

Round 1

Reviewer 1 Report

Excellent work. Congratulations.

Author Response

Reviewer 1 Comments

Excellent work. Congratulations.

Authors’ response

We thank the reviewer for the kind comment.

Reviewer 2 Report

Reviewer’s comments to Author:

1. The number of research subjects designed in this study is indeed relatively small.

2. What are the similarities between the results of this study and the results of previous studies? What features can we learn from the preliminary results of this study?

3. The results of various blood tests (such as Complete blood count) in the laboratory should provide normal values or reference intervals.

4. Is there any correlation or consistency between the laboratory test results of complicated appendicitis before the onset of the disease, the results of the ultrasound examination, and the results before and after the operation?

5. From the perspective of microbiology researchers, what are the limitations or obstacles of this study? Based on the current situation, is there a more rapid microbial detection method that can provide reference for clinical medication?

6. The preliminary results of this study are mostly complicated acute appendicitis caused by multiple microbial infections. What is the clinical suggestion for using antibiotics before and after surgery?

7. What are the relevant characteristics and learning objectives for our young physicians provided by the results of this study?

Author Response

Authors' responses to Reviewer's comments:

We thank the reviewer for the comments. We responded to each one individually, adding modifications in the manuscript in red color characters, accordingly.

  1. Reviewer’s comment

The number of research subjects designed in this study is indeed relatively small.

Authors’ response

We agree we the reviewer. This is why we mentioned the small study population number in the limitations paragraph [page 11, line 334]. Our study was conducted in a certain time period with an available population of a limited geographical area. Nevertheless, we referenced in the manuscript studies of analogous content with comparable numbers, i.e., Kroiča et al. (67 children), Guillet-Caruba et al. (93 children), Kadhim M.M. (54 children), Obinwa et al. (69 children), etc.

  1. Reviewer’s comment

What are the similarities between the results of this study and the results of previous studies? What features can we learn from the preliminary results of this study?

Authors’ response

We have compared the results of this study with those of previous research, as shown in the Discussion section [page 10, lines 294-299, page 11, lines 300-306]. However, according to the reviewers thoughts, we added more referenced research in the study, discussing further issues [characters in red color, page 11, lines 307-331].

The main features from the results of the study are that complicated appendicitis is encountered more when more than one organisms are implicated in the infection process. Furthermore, we gave a hint on Pseudomonas presence, which should be anticipated by preoperative targeted antibiotics in patients with clinical presentation implying complicated appendicitis. We outlined these outcomes in the discussion and conclusions sections extensively and also added a paragraph on this subject [characters in red cooler, page 11, 320-330]. We further outlined a possible synergistic collaboration of the microorganisms towards complicated appendicitis in the Conclusions section [characters in red color, page 11, lines 343-344].

  1. Reviewer’s comment

The results of various blood tests (such as Complete blood count) in the laboratory should provide normal values or reference intervals.

Authors’ response

We added the laboratory normal values of the blood tests to the information at the end of Table 1, with indexes in the Table [characters in red color, page 4, lines 158-161].

  1. Reviewer’s comment

Is there any correlation or consistency between the laboratory test results of complicated appendicitis before the onset of the disease, the results of the ultrasound examination, and the results before and after the operation?

Authors’ response

In this study we intended to focus strictly on the culture-based analysis outcomes. However, we have recently performed a study on the same population regarding the correlation between preoperative clinical and laboratory outcomes (Zachos, et al. Prediction of complicated appendicitis risk in children. Eur. Rev. Med. Pharmacol. Sci. 2021;25(23):7346-7353). A Classification and Regression Tree analysis (CRT) was used to create a multi-level classification algorithm based on the pediatric appendicitis score (PAS), neutrophils percentage, and the CRP. The model resulted in the prediction of  complicated appendicitis with 90% sensitivity and 78.6% specificity. The reviewer’s comment encourages the continuation of our research, by applying the outcomes of that study, and this which is under review, in a prospective model, adding ultrasonography into our equation.

  1. Reviewer’s comment

From the perspective of microbiology researchers, what are the limitations or obstacles of this study? Based on the current situation, is there a more rapid microbial detection method that can provide reference for clinical medication?

Authors’ response

There were no particular obstacles to this study. The microbial detection method we described in the manuscript was the most rapid available in our hospital at the moment. Particular attention was given to the direct inoculation of the clinical samples for the survival of oxygen-sensitive microorganisms such as anaerobes. The used culture media facilitated the growth of fastidious as well as anaerobic microorganisms. Based on the current situation, the molecular-based mass spectrometric identification method enabled faster and more efficiently the identification of microorganisms to the species level and, in many cases detected some enzymes, such as β-lactamase, offering an alternative approach to a phenotypic resistance testing.

We added this information to the manuscript in the Materials and Methods section, Microbiological assessment subsection [characters in red color, page 3, lines 113-119].

  1. Reviewer’s comment

The preliminary results of this study are mostly complicated acute appendicitis caused by multiple microbial infections. What is the clinical suggestion for using antibiotics before and after surgery?

Authors’ response

We added a paragraph on this clinical suggestion, extracted from our results in the discussion section (characters in red color, page 11, lines 320-330).

  1. Reviewer’s comment

What are the relevant characteristics and learning objectives for our young physicians provided by the results of this study?

Authors’ response

We added a paragraph to the Conclusions section, the last paragraph of the manuscript, as a consideration for young physicians (characters in red color, page 12, lines 349-360).

Reviewer 3 Report

The research is interesting and the results are of scientific significance. The minus of the Article is that the scientific basis for the research, especially in the Introduction, is based on outdated references. 10 to 15 years old and older references represent approximately 24% of all references.

Author Response

Reviewer’s comments

The research is interesting, and the results are of scientific significance. The minus of the Article is that the scientific basis for the research, especially in the Introduction, is based on outdated references. 10 to 15 years old and older references represent approximately 24% of all references.

Authors’ response

We thank the reviewer for the comment. To confront this problem, we searched and added in the discussion section the four most relevant to the topic publications during 2022 and 2023 in the PubMed database and added a paragraph on their outcomes as well [characters in red color, page 11, lines 307-319]. We also added two recent references in the introduction section.

Reviewer 4 Report

well done and interesting paper to read and to know about 

Author Response

Reviewer’s comment

Well done and interesting paper to read and to know about.

Authors’ response

We thank the reviewer for the kind comment.